# Effects of harvest number on the yield and quality of different alfalfa varieties under hydroponic conditions

Yang Bai[1,2,3], Wenlong Li[4], Meiying Liu●[1,2,3]*, Feng Li[4]*

1 College of Resources and Environment, Inner Mongolia Agricultural University, Inner Mongolia, China, 2 Inner Mongolia Key Laboratory of Soil Quality and Nutrient Resource, Hohhot, China, 3 Key Laboratory of Agricultural Ecological Security and Green Development at Universities of Inner Mongolia Autonomous, Hohhot, China, 4 Institute of Grassland Research of CAAS, Key Laboratory for Model Innovation in Forage Production Efficiency, Ministry of Agriculture and Rural Affairs, P. R. China

* liumeiyingimau@163.com (ML); lifeng03@caas.cn (FL)

## Abstract

Five alfalfa varieties, namely Zhongcao No.13, WL440HQ, WL525HQ, WL903, and WL712, were subjected to a hydroponic cultivation experiment under controlled conditions of 25℃ room temperature, a nutrient solution concentration of 1.5 times the standard Hoagland formulation, and a 10-hour photoperiod. The plants were harvested every 30 days for six consecutive harvests, and the effects of harvest time on the yield and quality of different alfalfa varieties were observed. Principal component and membership function analyses revealed that the acid detergent fiber (ADF), neutral detergent fiber (NDF), relative feeding value (RFV), and yield of WL903 ranged from 22.77% to 25.75%, 28.31% to 32.42%, 197.46 to 233.84 (unitless), and 0.41 kg·m$^{-2}$ to 0.59 kg·m$^{-2}$, respectively. The membership function analysis of the five alfalfa varieties indicated that the ranking based on D-value was WL903＞WL440HQ＞WL712＞WL525HQ＞Zhongcao No.13. Compared with the other four varieties, this variety was superior in terms of yield and relative feeding value and could serve as a candidate for regional hydroponic alfalfa cultivation. However, it should be noted that this result is based on specific light and nutrient solution conditions, and further cost-effectiveness verification is needed for large-scale applications.

## Introduction

Hydroponic cultivation is a technique in which plant roots are grown directly in a nutrient-rich solution. Hydroponic crops generally have a short growth cycle, a high multiple cropping index, high economic benefit, good product quality, no pollution, fast crop growth, no restrictions related to region and season, and are conducive to realizing industrialization and automation of production [1]. Today, hydroponics is widely applied in agricultural production [2]. In recent years, there has been vigorous

**Data availability statement:** All relevant data are within the manuscript and its Supporting information files.

**Funding:** This work was supported by the "Technology Xingmeng" Action Key Projects (NMKJXM202307-01), the Central Public-interest Scientific Institution Basal Research Fund (Y2025CY21), the National Dairy Technology Innovation Center Project (2022SR-1), and the Project of Technology Plan in Inner Mongolia (2025YFHH0270). The funding sources provided the research facilities and equipment necessary for this study.

**Competing interests:** The authors have declared that no competing interests exist.

development of hydroponic production systems globally. Hydroponic forage has the advantages of a small footprint, low water demand, short production cycle, little adverse effects of natural conditions, good palatability, high digestibility, and clean production throughout the year [3]. China, as the world's most populous nation, has effectively ensured food security for approximately 20% of the global population using only 9% of the world's arable land and 6% of its freshwater resources [4,5]. Hydroponic cultivation has great development space and potential in China. As a new and efficient production system, at present, the application of hydroponic cultivation is relatively limited to scientific and technological demonstration, observation, and guidance; due to the lack of technical standardization, there is still a long way to go to achieve large-scale hydroponic production.

Alfalfa (*Medicago sativa*) is recognized as the "king of forages" [6] due to its significant role in the agricultural economy [7]. The majority of global alfalfa production, around 60%, is concentrated in four key countries: the United States and Canada in North America, Argentina in South America, and China in East Asia. Alfalfa cultivation predominantly occurs in regions with a temperate climate [8]. Since 2014, China's alfalfa imports have exceeded 1 million tons, surpassing those of South Korea; China has become the world's second-largest importer of alfalfa. From 2008 to 2016, the proportion of alfalfa imports increased from only 0.47% to 24.30% [9]. After 2017, China's alfalfa imports gradually stabilized. High-quality forage varieties introduced from abroad play a vital role in the development of China's dairy industry. Alfalfa is a leguminous forage with good palatability, a high crude protein content, and a low fiber content. All kinds of livestock and poultry like to feed, which is a good source of protein and vitamins [10]. Fresh grass or silage can increase milk production and play an important role in the safe production of the dairy industry [11]. Given its advantages, including the small land area required for production and the short production cycle, hydroponic technology has become a potential solution to address limitations in cultivated land area. However, existing studies have focused on single varieties or single harvests and lack systematic comparisons of multiple varieties at high harvest numbers. In this study, five main alfalfa varieties were grown under hydroponic conditions and harvested six times in succession. The yield and quality of the five alfalfa varieties were compared. The results of this study provide insight into alfalfa varieties suitable for hydroponic systems, providing a reference for forage production in resource-constrained areas.

## Experiments and methods

### Test materials

The alfalfa varieties used in this experiment are shown in Table 1.

### Experimental design

Five alfalfa varieties, namely Zhongcao No. 13, WL440HQ, WL525HQ, WL903, and WL712, were selected for this experiment. Zhongcao No. 13 is a Chinese-origin alfalfa variety characterized by stress tolerance and harvest resistance, while the "WL" series

**Table 1. Test material sources and details.**

| Number | Variety | Source | Merit |
|---|---|---|---|
| 1 | Zhongcao No.13 | Institute of Grassland Research of Caas | Strong stress tolerance and harvest tolerance |
| 2 | WL440HQ | Beijing Zhengdao Seed Industry Co., Ltd | High yield characteristics, good quality. |
| 3 | WL525HQ | Beijing Zhengdao Seed Industry Co., Ltd | |
| 4 | WL903 | Beijing Zhengdao Seed Industry Co., Ltd | |
| 5 | WL712 | Beijing Zhengdao Seed Industry Co., Ltd | |

are American-origin alfalfa varieties known for high yield and quality. The hydroponic system utilized the shallow flow technique. The height of the water was 2 cm. The tray size was 1.2 m × 0.9 m × 0.2 m. The planting density was 104 plants/m². Each variety was repeated in triplicate. The room temperature was maintained at 25°C. The concentration of Hoagland nutrient solution was 1.5 times that of standard Hoagland solution (the formula for standard Hoagland nutrient solution is: $Ca(NO_3)_2 \cdot 4H_2O$ 945 mg/L, $KNO_3$ 505 mg/L, $NH_4H_2PO_4$ 115 mg/L, $MgSO_4 \cdot 7H_2O$ 493 mg/L). The pH was maintained at 5.8 ± 0.2. The conductivity was 2.0 ± 0.2 mS/cm. An LED white light source was used for illumination with an intensity of 400 µmol·m$^{-2}$·s$^{-1}$. The light period was 8:00–11:30 (3.5h), 15:00–18:30 (3.5h), and 20:00–23:00 (3h); the dark period was 14 h. This light schedule was determined based on the results of preliminary experiments. Six successive harvests were carried out, with one harvest performed every 30 days (bud stage), to observe the effect of different alfalfa varieties on quality and yield under hydroponic conditions.

## Measurement indicators and methods

**Determination of the yield index.** Yield index determination: Harvesting was performed at a 5 cm root length after 30 days of growth. The fresh mass was immediately weighed and then deactivated at 105°C for 5 min in an oven and dried at 65°C to a constant mass to prepare for subsequent yield index determination. The dry matter weight (DM) was calculated according to the moisture determination method [12], and the hay yield (kg·m$^{-2}$) was calculated.

The hay yield calculation formula was as follows:

$$\text{Hay yield} = \text{fresh yield} \times (1 - \text{moisture content}) \tag{1}$$

**Determination of the nutrient content.** Alfalfa hay samples were crushed through a 40-mesh sieve in order to determine the nutritional quality traits. The crude protein (CP) content was determined by the Kjeldahl method [13], the neutral detergent fiber (NDF) content was determined according to the GB/T20806-2022 standard [14], and the acid detergent fiber (ADF) content was determined according to the NY/T1459-2007 standard [15]

The relative feeding value (RFV) was calculated as follows:

$$\text{RFV} = \frac{\text{DMI} \times \text{DDM}}{1.29} \tag{2}$$

In the formula, DMI (dry matter intake) is the random feed intake (% BW) of dry matter in the roughage of livestock per unit weight; DMI (digestible dry matter) is the proportion of digestible dry matter to total dry matter (% DM). The prediction models for DMI and DDM were as follows:

$$\text{DMI} = \frac{120}{\text{NDF}} \tag{3}$$

$$\text{DDM} = 88.9 - 0.779 \times \text{ADF} \tag{4}$$

## Statistical analysis

Repeated measures analysis of variance was performed using IBM SPSS Statistics 27. The spherical symmetry test (Mauchly's test) was performed first, i.e., to check that the covariance matrix of repeated measures error was proportional to the identity matrix after orthogonal contrast transformation, to evaluate the suitability of the data. If the test results met the assumptions, no correction was applied; otherwise, the Greenhouse-Geisser correction was applied. Then, the analysis of variance results for the interaction between alfalfa variety and harvest time were judged. The Tukey method was used for multiple comparisons. $p < 0.05$ was considered statistically significant. Correlations, principal component analysis, and plot analysis were performed using Lab Origin 2021. Based on fuzzy mathematics principles, a membership function evaluation system was constructed, and the CP, DMI, DDM, and other key indices were integrated to achieve quantitative ranking of the comprehensive production performance of the alfalfa varieties. The membership function was as follows: $\mu(Xi)=(Xi-Xmin)/(Xmax-Xmin)$, where $Xi$ represents the ith comprehensive index value, $Xmin$ represents the minimum value of the ith comprehensive index, $Xmax$ represents the maximum value of the ith comprehensive index, and $\mu(Xi)$ represents the membership function value of the ith index. The weight of each index was calculated using the membership function value of the comprehensive index, and then the D value of the comprehensive evaluation of each variety was calculated [16].

## Results and analysis

### Effects of alfalfa variety on nutritional quality and yield traits under hydroponic conditions

The results of the repeated measures analysis of variance of the nutrient quality and yield of the five alfalfa varieties harvested at six successive time points are shown in Table 2. Variety significantly affected the ADF, NDF, DMI, DDM, RFV, CP, yield, and leaf:stem ratio ($p < 0.001$), successive harvest significantly affected the ADF, NDF, DMI, DDM, RFV, CP, yield, and leaf:stem ratio ($p < 0.001$), and their interaction significantly affected the nutritional quality and yield ($p < 0.001$). The RFV values of the five alfalfa varieties ranged from 148.10 to 233.84 after six consecutive harvests; after the 2nd to 6th harvests, WL903 had a significantly higher RFV content (ranging from 211.34 to 233.84) than the other four varieties ($p < 0.001$); WL903, WL440HQ, WL712, and WL525HQ all reached the standard of special alfalfa and Zhongcao No. 13 reached the standard of general to good alfalfa, according to the "American alfalfa hay quality inspection index and classification guide" issued by the United States Department of Agriculture [17]. The CP content differed significantly among the five alfalfa varieties ($p < 0.001$), with the values ranging from 22.31–35.27%. The results of the repeated measures analysis of variance showed that the CP content of WL440HQ exhibited a significant advantage under three to five consecutive harvests ($p < 0.001$), with CP contents of 34.43%, 35.27%, and 33.69%, respectively. The CP content of Zhongcao No. 13 was significantly lower than that of the other four varieties under six consecutive harvests. The ADF values of the five varieties ranged from 22.77% to 35.44% under six consecutive harvests, and the ADF content of WL903 was significantly lower than that of the other four varieties ($p < 0.001$), ranging from 22.77% to 25.24% under two to six consecutive harvests. The NDF values of the five varieties ranged from 28.31% to 38.50% under six consecutive harvests, and WL903 had a significantly lower NDF content ($p < 0.001$) than the other varieties (28.31–30.12%) under three to six consecutive harvests. The DMI values of the five alfalfa varieties ranged from 3.12% to 4.24% under six consecutive harvests, among which, the DMI content of WL903 was significantly higher than that of the other varieties ($p < 0.001$) at three to six consecutive harvests (3.98–4.24%). The DDM values of the five alfalfa varieties under six consecutive harvests ranged from 61.29% to 71.16%, among which, WL903 had a significantly higher DDM content ($p < 0.001$) than the other varieties under two to six consecutive harvests (69.24–71.16%). The leaf:stem ratio of the five alfalfa varieties ranged from 0.56% to 1.17% under six consecutive harvests. WL712 had a significantly higher leaf:stem ratio than the other varieties ($p < 0.001$), ranging from 1.05% to 1.17% under two to four consecutive harvests. The hay yield of the five alfalfa varieties ranged from 0.11 kg·m$^{-2}$ to 0.59 kg·m$^{-2}$ under six consecutive harvests. The yield of WL903 under one to six consecutive harvests was significantly higher than that of the other varieties ($p < 0.001$), and the yield per unit area ranged from 0.41 kg·m$^{-2}$

**Table 2. Repeated measures analysis of variance of the nutritional quality and yield traits of five alfalfa varieties over six consecutive harvests.**

| ADF% | Zhongcao No.13 | WL440HQ | WL525HQ | WL903 | WL712 | F | P |
|---|---|---|---|---|---|---|---|
| 1 | 35.44±0.07 | 27.15±0.01A | 28.62±0.03AB | 25.75±0.02ABC | 26.28±0.47ABC | 1,031.682 | <0.001 |
| 2 | 34.73±0.26a | 26.22±0.18aA | 29.26±0.14aAB | 24.88±0.03aABC | 25.93±0.07ACD | 1,974.970 | <0.001 |
| 3 | 34.22±0.02ab | 26.49±0.10A | 28.29±0.10bAB | 25.24±0.19ABC | 25.77±0.23ABCD | 1,887.895 | <0.001 |
| 4 | 34.66±0.04a | 26.99±0.06bA | 26.15±0.14abcAB | 24.36±0.09abcABC | 26.02±0.32ABD | 1,876.020 | <0.001 |
| 5 | 33.22±0.20abcd | 24.91±0.07abcdA | 28.08±0.03abdAB | 23.21±0.19abcdABC | 26.11±0.28ABCD | 1,421.541 | <0.001 |
| 6 | 29.47±0.12abcde | 24.58±0.30abcdA | 26.05±0.12abceAB | 22.77±0.03abcdABC | 24.42±0.07abcdeACD | 807.335 | <0.001 |
| F | 1817.201 | 1778.032 | 393.800 | 2332.565 | 72.271 | | |
| P | <0.001 | <0.001 | <0.001 | <0.001 | <0.001 | | |
| Whole-test: groups (F,P) 8,858.466, <0.001; stubbles (F,P) 660.194, <0.001; interaction (F,P)77.945, <0.001 | | | | | | | |
| NDF% | Zhongcao No.13 | WL440HQ | WL525HQ | WL903 | WL712 | F | P |
| 1 | 38.50±0.09 | 31.13±0.05A | 31.53±0.26A | 32.43±0.20ABC | 31.84±0.05ABD | 1,168.804 | <0.001 |
| 2 | 37.60±0.11a | 30.47±0.12aA | 31.65±0.14AB | 30.60±0.13aAC | 30.92±0.08aABC | 2,061.830 | <0.001 |
| 3 | 37.33±0.12ab | 30.86±0.03abA | 30.79±0.25abA | 30.12±0.09abABC | 30.68±0.07abAD | 1,558.057 | <0.001 |
| 4 | 37.14±0.33a | 31.17±0.24bA | 30.81±0.08abA | 29.06±0.04abcABC | 31.47±0.22cACD | 640.152 | <0.001 |
| 5 | 36.30±0.07abcd | 29.61±0.25abcdA | 31.01±0.00AB | 29.00±0.12abcAC | 31.97±0.69bcABD | 227.631 | <0.001 |
| 6 | 35.75±0.22abcd | 29.09±0.03abcdA | 29.80±0.08abcdeAB | 28.31±0.13abcdeABC | 29.52±0.12abcdeABD | 1,573.926 | <0.001 |
| F | 140.422 | 170.756 | 117.211 | 311.360 | 208.077 | | |
| P | <0.001 | <0.001 | <0.001 | <0.001 | <0.001 | | |
| Whole-test: groups (F,P) 2,669.359, <0.001; stubbles (F,P)324.558, <0.001; interaction (F,P)34.457, <0.001 | | | | | | | |
| DMI% | Zhongcao No.13 | WL440HQ | WL525HQ | WL903 | WL712 | F | P |
| 1 | 3.12±0.01 | 3.86±0.01A | 3.81±0.04A | 3.70±0.02ABC | 3.77±0.01ABD | 767.917 | <0.001 |
| 2 | 3.19±0.01a | 3.94±0.02aA | 3.79±0.02AB | 3.92±0.02aAC | 3.88±0.01aABCD | 1,395.031 | <0.001 |
| 3 | 3.21±0.01a | 3.89±0.00abA | 3.90±0.03abA | 3.98±0.02abABC | 3.91±0.01abAD | 1,152.100 | <0.001 |
| 4 | 3.23±0.03a | 3.85±0.03bA | 3.89±0.01abA | 4.13±0.01abcABC | 3.81±0.03cACD | 624.094 | <0.001 |
| 5 | 3.31±0.01ab | 4.05±0.04abcdA | 3.87±0.00AB | 4.14±0.02abcAC | 3.75±0.09bcABD | 179.996 | <0.001 |
| 6 | 3.36±0.02abcd | 4.13±0.01abcdA | 4.03±0.01abcdeAB | 4.24±0.02abcdeABC | 4.06±0.02abcdeABD | 1,548.243 | <0.001 |
| F | 91.494 | 158.703 | 203.448 | 473.857 | 202.241 | | |
| P | <0.001 | <0.001 | <0.001 | <0.001 | <0.001 | | |
| Whole-test: groups (F,P)1,965.572, <0.001; stubbles (F,P)324.067, <0.001; interaction (F,P)38.061, <0.001 | | | | | | | |
| DDM% | Zhongcao No.13 | WL440HQ | WL525HQ | WL903 | WL712 | F | P |
| 1 | 61.29±0.06 | 67.75±0.01A | 66.60±0.02AB | 68.84±0.01ABC | 68.43±0.36ABC | 1,064.689 | <0.001 |
| 2 | 61.84±0.20a | 68.48±0.14aA | 66.11±0.11aAB | 69.52±0.02aABC | 68.70±0.06ACD | 1,933.735 | <0.001 |
| 3 | 62.25±0.02ab | 68.26±0.08A | 66.86±0.08bAB | 69.24±0.15ABC | 68.83±0.18ABCD | 1,862.751 | <0.001 |
| 4 | 61.90±0.03a | 67.87±0.05bA | 68.53±0.11abcAB | 69.93±0.08abcABC | 68.63±0.25ABD | 1,865.670 | <0.001 |
| 5 | 63.02±0.15abcd | 69.50±0.06abcdA | 67.02±0.03abdAB | 70.82±0.15abcdABC | 68.56±0.22ABCD | 1,411.514 | <0.001 |
| 6 | 65.94±0.09abcde | 69.75±0.24abcdA | 68.61±0.10abceAB | 71.16±0.02abcdABC | 69.88±0.06abcdeACD | 767.040 | <0.001 |
| F | 1788.236 | 1804.646 | 417.635 | 2445.483 | 68.152 | | |
| P | <0.001 | <0.001 | <0.001 | <0.001 | <0.001 | | |
| Whole-test: groups (F,P) 9,036.154, <0.001; stubbles (F,P) 653.481, <0.001; interaction (F,P) 77.220, <0.001 | | | | | | | |
| RFV | Zhongcao No.13 | WL440HQ | WL525HQ | WL903 | WL712 | F | P |
| 1 | 148.10±0.48 | 202.47±0.29A | 196.51±1.71AB | 197.46±1.26AB | 199.91±1.40 AC | 1,166.334 | <0.001 |
| 2 | 153.01±0.92a | 209.06±1.22aA | 194.31±1.18aAB | 211.34±0.97aAC | 206.70±0.66aACD | 1,750.619 | <0.001 |

*(Continued)*

| ADF% | Zhongcao No.13 | WL440HQ | WL525HQ | WL903 | WL712 | F | P |
|---|---|---|---|---|---|---|---|
| 3 | 155.11±0.45a | 205.75±0.37abA | 202.02±1.82abAB | 213.85±0.20abABC | 208.64±0.97aABCD | 1,849.366 | <0.001 |
| 4 | 155.05±1.28a | 202.59±1.68bA | 206.90±0.83abcAB | 223.86±0.08abcABC | 202.88±2.15cAD | 1,015.583 | <0.001 |
| 5 | 161.46±0.09abd | 218.31±1.61abcdA | 201.05±0.09bdAB | 227.19±1.41abcABC | 199.55±4.91bcABD | 333.455 | <0.001 |
| 6 | 171.59±1.30abcde | 223.07±0.53abcdA | 214.16±0.25abcdeAB | 233.84±0.99abcdeABC | 220.15±0.69abcdeABCD | 2,474.225 | <0.001 |
| F | 635.071 | 1164.377 | 108.721 | 529.146 | 837.690 | | |
| P | <0.001 | <0.001 | <0.001 | <0.001 | <0.001 | | |

Whole-test: groups (F,P) 3,073.191, <0.001; stubbles (F,P) 564.650, <0.001; interaction (F,P) 49.174, <0.001

| CP% | Zhongcao No.13 | WL440HQ | WL525HQ | WL903 | WL712 | F | P |
|---|---|---|---|---|---|---|---|
| 1 | 25.85±0.07 | 33.41±0.14A | 31.89±0.06AB | 28.09±0.05ABC | 33.46±0.01ACD | 5,767.380 | <0.001 |
| 2 | 22.76±0.07a | 32.57±0.03aA | 34.06±0.03aAB | 30.49±0.08aABC | 32.89±0.02aABCD | 25,260.920 | <0.001 |
| 3 | 22.31±0.30a | 34.43±0.08abA | 33.50±0.33abAB | 31.26±0.01abABC | 32.40±0.18aABCD | 1,518.757 | <0.001 |
| 4 | 22.50±0.47a | 35.27±0.25abcA | 32.44±0.14bcAB | 30.67±0.03aABC | 31.25±0.04abcABC | 1,125.220 | <0.001 |
| 5 | 23.87±0.29abcd | 33.69±0.03bcdA | 32.03±0.09bcAB | 29.38±0.10abcdABC | 31.63±0.20abcABD | 1,512.047 | <0.001 |
| 6 | 25.81±0.13bcde | 32.46±0.08acdeA | 34.28±0.02acdeAB | 31.12±0.14abeABC | 33.55±0.19bcdeABCD | 2,108.863 | <0.001 |
| F | 502.182 | 148.586 | 497.028 | 720.389 | 122.250 | | |
| P | <0.001 | <0.001 | <0.001 | <0.001 | <0.001 | | |

Whole-test: groups (F,P) 19,336.357, <0.001; stubbles (F,P) 102.262, <0.001; interaction (F,P) 161.645, <0.001

| Hay yield kg·m$^{-2}$ | Zhongcao No.13 | WL440HQ | WL525HQ | WL903 | WL712 | F | P |
|---|---|---|---|---|---|---|---|
| 1 | 0.38±0.00 | 0.47±0.01A | 0.15±0.00AB | 0.54±0.01ABC | 0.22±0.00ABCD | 6,065.750 | <0.001 |
| 2 | 0.28±0.00a | 0.31±0.01aA | 0.12±0.00aAB | 0.59±0.01aABC | 0.26±0.01aBCD | 2,169.333 | <0.001 |
| 3 | 0.24±0.01ab | 0.37±0.00abA | 0.11±0.01aAB | 0.44±0.01abABC | 0.30±0.01abABCD | 1,181.083 | <0.001 |
| 4 | 0.27±0.00ac | 0.31±0.01acA | 0.16±0.00bcAB | 0.41±0.01abcABC | 0.26±0.00acBCD | 1,802.000 | <0.001 |
| 5 | 0.31±0.01abcd | 0.33±0.00acd | 0.19±0.00abcdAB | 0.57±0.01acdABC | 0.33±0.01abcdCD | 1,714.700 | <0.001 |
| 6 | 0.30±0.00abcd | 0.42±0.01abcdeA | 0.22±0.00abcdeAB | 0.41±0.00abceABC | 0.39±0.00abcdeABCD | 3,371.500 | <0.001 |
| F | 306.632 | 633.901 | 127.657 | 790.075 | 612.365 | | |
| P | <0.001 | <0.001 | <0.001 | <0.001 | <0.001 | | |

Whole-test: groups (F,P) 6,665.097, <0.001; stubbles (F,P) 677.052, <0.001; interaction (F,P) 487.026, <0.001

| leaf-stem ratio % | Zhongcao No.13 | WL440HQ | WL525HQ | WL903 | WL712 | F | P |
|---|---|---|---|---|---|---|---|
| 1 | 0.56±0.01 | 0.78±0.01A | 0.80±0.03A | 0.84±0.01AB | 0.83±0.01AB | 203.200 | <0.001 |
| 2 | 0.71±0.01a | 0.80±0.04A | 0.90±0.00aAB | 0.91±0.01aAB | 1.17±0.03aABCD | 171.904 | <0.001 |
| 3 | 0.73±0.01a | 0.89±0.02abA | 0.94±0.01aAB | 0.79±0.00abABC | 1.05±0.02abABCD | 286.240 | <0.001 |
| 4 | 0.72±0.00a | 0.80±0.01cA | 0.94±0.01aAB | 1.01±0.01abcABC | 1.14±0.02acABCD | 472.731 | <0.001 |
| 5 | 0.72±0.02a | 1.08±0.01abcdA | 1.15±0.02abcdAB | 0.98±0.02abcABC | 1.14±0.01acABD | 415.971 | <0.001 |
| 6 | 0.69±0.02ac | 1.07±0.02abcdA | 1.06±0.02abcdeA | 0.82±0.02bdeABC | 1.03±0.01abdeAD | 312.619 | <0.001 |
| F | 73.634 | 207.335 | 179.312 | 80.492 | 232.702 | | |
| P | <0.001 | <0.001 | <0.001 | <0.001 | <0.001 | | |

Whole-test: groups (F,P) 776.097, <0.001; stubbles (F,P) 442.847, <0.001; interaction (F,P) 92.705, <0.001

Note: The data were analyzed using repeated-measures ANOVA. Within each data cell: a = $p < 0.05$ compared with the first crop; b = $p < 0.05$ compared with the second crop; c = $p < 0.05$ compared with the third crop; d = $p < 0.05$ compared with the fourth crop; e = $p < 0.05$ compared with the fifth crop; A = $p < 0.05$ compared with Zhongcao No. 13; B = $p < 0.05$ compared with WL440H; C = $p < 0.05$ compared with WL525HQ; D = $p < 0.05$ compared with WL903. All pairwise comparisons were Bonferroni corrected.

to 0.59 kg·m⁻². WL525HQ had the lowest yield under one to six consecutive harvests, ranging from 0.11 kg·m⁻² to 0.22 kg·m⁻².

## Impact of harvest timing on the nutritional quality and yield characteristics of alfalfa in a hydroponic system

According to Table 2, the RFV content of WL903 ranged from 197.46 to 233.84 under six consecutive harvests. The RFV content reached 233.84 in the sixth harvest, which was significantly different from that of the first five harvests. The CP content ranged from 28.09% to 31.26%. The CP content of the first crop was 28.09%, and there was a significant difference ($p < 0.001$) between the first and the last five crops. The hay yield ranged from 0.41 kg·m⁻² to 0.59 kg·m⁻² under six consecutive harvests. The yields of the second and fifth harvests showed significant superiority ($p < 0.001$), being 0.59 kg·m⁻² and 0.57 kg·m⁻², respectively, while the lowest yields, in the fourth and sixth harvests, were 0.41 kg·m⁻², respectively.

## The impact of harvesting frequency on the leaf-to-stem ratio and yield of various alfalfa varieties

It can be seen in Fig 1a that the leaf:stem ratio of Zhongcao No. 13 was lower than that of the other varieties across the six harvest times. Overall, the leaf:stem ratio of WL712 was higher than those of the other varieties. Overall, the leaf-to-stem ratios of the five varieties first increased and then decreased with successive harvests. In Fig 1b, it can be seen that the yield of WL903 was significantly higher than that of the other four varieties in harvests one to five, and the yield of WL440HQ was higher than that of WL903 in the sixth harvest. Overall, WL903 exhibited a declining yield with more harvests, whereas WL712 and WL525HQ showed increasing yields. In summary, WL903 exhibited the highest yield among the five varieties, with WL525HQ yielding the least across the six harvests.

## Effects of harvest time on the nutritional quality traits of different alfalfa varieties

From Fig 2, it can be seen that Zhongcao No. 13 had the highest ADF content, WL903 had the lowest ADF content, Zhongcao No. 13 had the highest NDF content, WL903 had the lowest NDF content in three harvests, WL903 had the

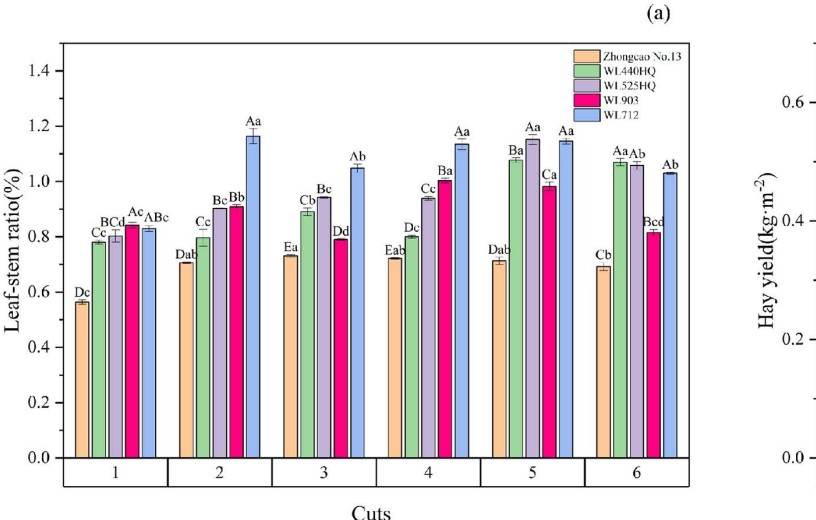 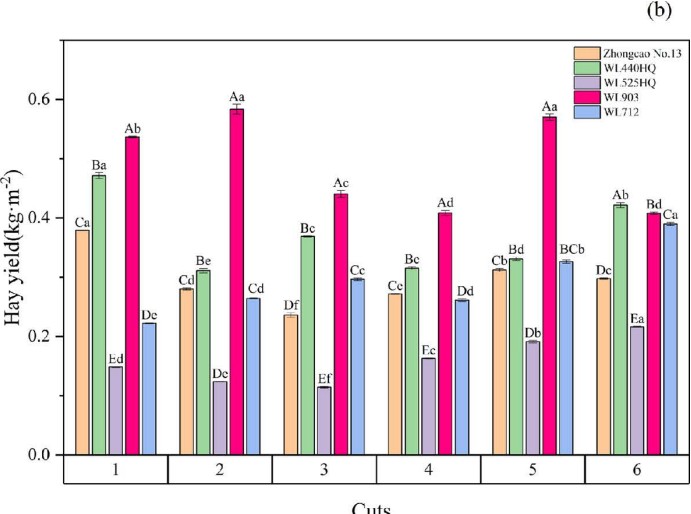

**Fig 1. Yield and leaf-to-stem ratio of different alfalfa varieties over six consecutive harvests.** Note: A indicates a significant difference between different varieties at the same harvesting time (p<0.05); a indicates a significant difference between different harvesting times for the same variety (p<0.05).

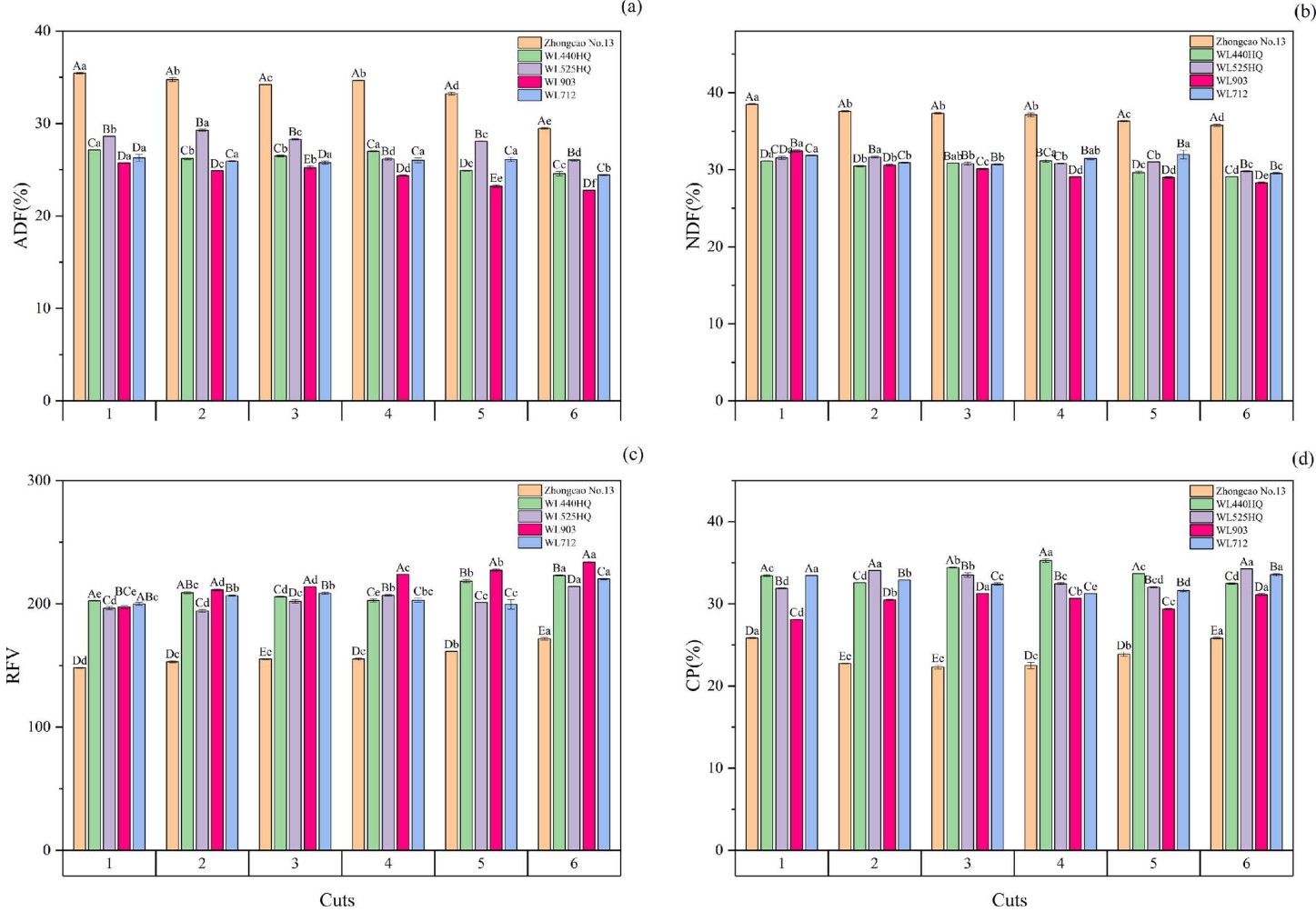

**Fig 2. Nutritional quality of different alfalfa varieties over six consecutive harvests.** Note: (a) ADF content; (b) NDF content; (c) RFV; (d) CP content.

highest ADF content in three harvests, and WL903 had the lowest ADF content in three harvests. Over six consecutive harvests, Zhongcao No.13 consistently had the highest ADF and NDF contents. WL903 had the lowest ADF content (22.77–25.24%) from the second to the sixth harvest, and the lowest NDF content (28.31–30.12%) from the third to the sixth harvest. The RFVs of Zhongcao No. 13 and WL903 increased with increases in the number of harvests. The RFV of Zhongcao No. 13 was the lowest over six consecutive harvests, while the RFV of WL903 was higher than the other four varieties. The CP contents of WL440HQ, WL525HQ, and WL712 were higher than those of Zhongcao No. 13 and WL903, but the CP content of Zhongcao No. 13 was the lowest among the five varieties.

## Analysis of relationship

In order to further analyze the relationships between different traits, the correlations between alfalfa yield and nutritional quality traits were analyzed, as shown in Fig 3. RFV was positively correlated with CP, DMI, DDM, and the leaf:stem ratio ($p < 0.01$), DM was negatively correlated with RFV, CP, DMI, DDM, and the leaf:stem ratio ($p < 0.01$), CP was positively

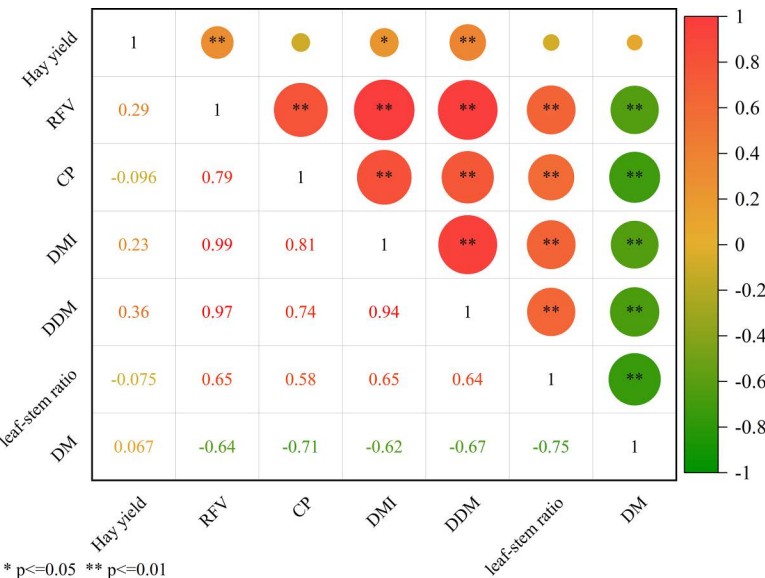

**Fig 3. Correlation coefficient matrix between yield and nutritional quality traits of alfalfa.** Note: * and * * represent significance at the 0.05 and 0.01 levels, respectively.

correlated with DMI, DDM, and the leaf:stem ratio ($p<0.01$), hay yield was positively correlated with RFV and DDM ($p<0.01$), and hay yield was positively correlated with DMI ($p<0.05$). DMI and DDM were positively correlated with the leaf:stem ratio, and DMI was correlated with DDM ($p<0.01$).

## Principal component analysis

Principal component analysis (PCA) was carried out for Hay yield、RFV、CP、DMI、DDM、leaf-stem ratio、DM. According to the results shown in Table 3 and Fig 4, the contribution rates of the first two principal components were 68.073% and 18.971%, respectively, and the cumulative contribution rate reached 87.043%. According to the principle that the cumulative contribution rate of principal components should be greater than 85%, the first two principal components were extracted to represent the information of the original seven indices. The eigenvalues of the first two principal

**Table 3. Loading matrix of principal components.**

| Project | Principal component analysis | |
|---|---|---|
| | 1 | 2 |
| Hay yield | 0.070 | 0.809 |
| RFV | 0.439 | 0.188 |
| CP | 0.401 | −0.193 |
| DMI | 0.436 | 0.145 |
| DDM | 0.432 | 0.234 |
| leaf-stem ratio | 0.360 | −0.262 |
| DM | −0.368 | 0.358 |
| Eigenvalue | 4.765 | 1.328 |
| Contribution rate (%) | 68.073 | 18.971 |
| The contribution rate (%) | 68.073 | 87.043 |

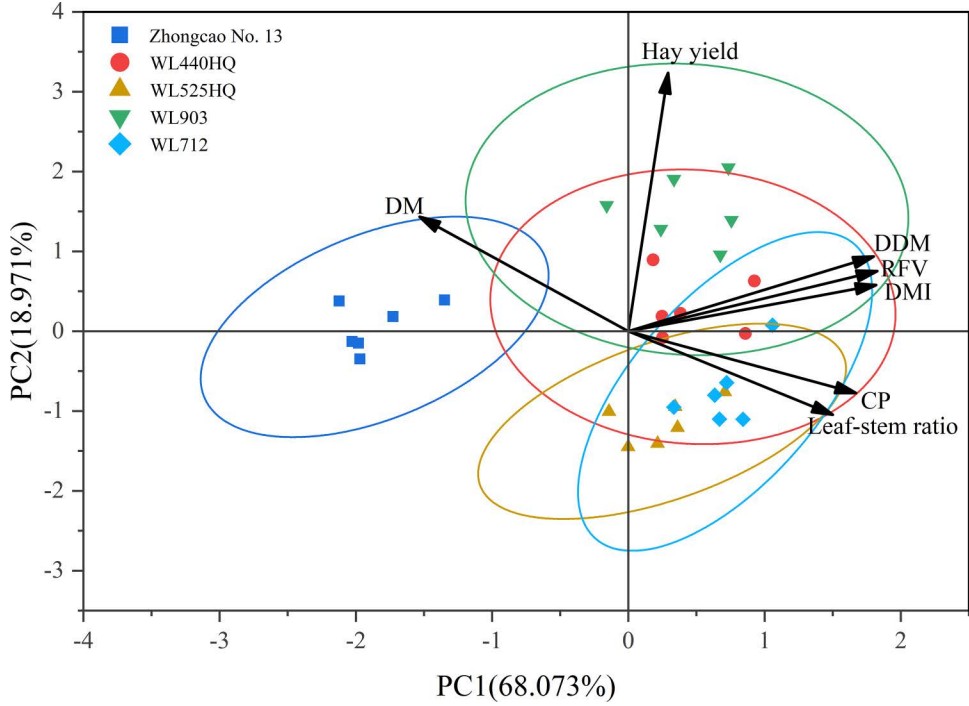

**Fig 4. Principal component score plot.**

components were 4.765 and 1.328, respectively. Notably, the first principal component, with an eigenvalue of 4.765, captured 68.073% of the variability in alfalfa biomass and nutritional quality traits. The corresponding eigenvectors with higher loadings included RFV, CP, DMI, DDM, and the leaf:stem ratio; this factor predominantly reflected a "nutritional quality traits" factor of alfalfa. In contrast, the second principal component was primarily characterized by yield traits, reflecting a "biomass" factor of alfalfa.

## Assessment of D-values and comprehensive evaluation of diverse alfalfa varieties across six harvest times

According to the measured values and weights of each single index for the five alfalfa varieties, the comprehensive index value F (Xi), membership function value μ(Xi), and D value of each variety were calculated, and the D values were ranked to compare the yield and quality indices of the five varieties of alfalfa under six successive harvests. The higher the D value, the better the yield and nutritional quality of the alfalfa variety under the harvest and hydroponic conditions. The results in Table 4 show that the D value of WL903 was the highest (0.813), while the D value of Zhongcao No. 13 was the lowest (0.157).

**Table 4. Comprehensive evaluation of the membership functions of five different varieties of alfalfa.**

| Cuts | F(X1) | F(X2) | μ(X1) | μ(X2) | D-value | Comprehensive evaluation and ranking |
|---|---|---|---|---|---|---|
| Zhongcao No.13 | −1.862 | 0.053 | 0.082 | 0.429 | 0.157 | 5 |
| WL440HQ | 0.474 | 0.305 | 0.816 | 0.501 | 0.748 | 2 |
| WL525HQ | 0.248 | −1.129 | 0.745 | 0.091 | 0.603 | 4 |
| WL903 | 0.431 | 1.526 | 0.803 | 0.851 | 0.813 | 1 |
| WL712 | 0.710 | −0.755 | 0.890 | 0.198 | 0.740 | 3 |

## Discussion

### Differences in the yield and quality of alfalfa among different varieties

The correlation analysis showed that CP was significantly positively correlated with the leaf-to-stem ratio ($p < 0.01$). This is consistent with previous research results [18]. This finding may be related to the higher CP content in alfalfa leaves, thus resulting in a significant positive correlation between the leaf-to-stem ratio and CP ($p < 0.01$). From the results of the principal component analysis, RFV, CP, DMI, DDM, and the leaf-to-stem ratio were higher and positive in the first principal component eigenvector, indicating that the higher the DMI and DDM, the higher the RFV value. The larger and positive value of the second principal component eigenvector was hay yield, which is basically consistent with previous results [19]. The growth and development of alfalfa are not only restricted by its own genetic characteristics [20] but also by many external factors such as temperature and light conditions [21]. This study identified substantial variations in yield and quality across five different alfalfa varieties. WL903 exhibited excellent yield (0.41–0.59 kg·m$^{-2}$) over six successive harvests. Compared with the other varieties, WL903 had the lowest ADF and NDF contents (22.77–25.75% and 28.31–32.43%, respectively). WL440HQ had a better CP content; the highest CP content was 35.27%, and the yield was second only to WL903. From the comprehensive evaluation of the subordinate function, WL903 was superior in terms of yield. These findings indicate that genetic characteristics and physiological adaptability are important factors determining yield and quality.

### Effects of harvest time on the yield and quality of alfalfa

Previous studies have demonstrated that the yield and nutritional quality of alfalfa are not only influenced by the genetic characteristics of the variety but are also associated with the frequency of harvest [22]. With an increase in the number of harvests, the yield and leaf-to-stem ratio of the WL903 variety increased. When the yield increases, the leaf-to-stem ratio will also increase; the number of alfalfa branches in the stubble increases, and the number of leaves also increases, which promotes an increase in yield and the leaf-to-stem ratio, and vice versa when the yield decreases. However, the relative feeding value of WL903 increased with increases in the number of harvests. This is opposite to the overall yield, indicating that the yield value of alfalfa may be sacrificed while pursuing quality. This is consistent with existing research results [23]. The yields of the remaining four varieties exhibited a rising trend with each subsequent harvest after the third, aligning with the increasing leaf-to-stem ratio observed. This suggests that moderate harvest enhances alfalfa's branching and regenerative capacity, consequently boosting its leaf-to-stem ratio. Among the six consecutive harvests, the hay yield of the first harvest of the five varieties ranged from 0.15 kg·m$^{-2}$ to 0.54 kg·m$^{-2}$, higher than that of the other five harvests. This is consistent with previous research results [24].

### Effects of hydroponic conditions on the growth of alfalfa

Hydroponic conditions provide a stable environment for alfalfa growth, avoid the interference of soil factors, and are conducive to studying the independent effects of harvest and variety on alfalfa. In this study, alfalfa grew rapidly in hydroponic culture and could be harvested once a month. The CP content of most alfalfa varieties was above 30%, much higher than that of field alfalfa. This high CP content may be related to good nutrient supply and early harvest afforded by the hydroponic environment. The CP content of alfalfa harvested at the 10% flowering period in a conventional field is usually 20–25% [25]. It has been reported that the CP content of alfalfa harvested in 15 days can reach more than 40% [26]. It has also been found that 10% alfalfa fresh grass can improve the growth performance and nutrient digestibility of beef cattle [27]. If hydroponic alfalfa is added to the diet, corresponding adjustments should be made according to the different nutritional quality traits of hydroponic alfalfa.

## Conclusions

The results showed that WL903 had a significant agronomic advantage and superior nutritional characteristics compared with the other four varieties in the hydroponic culture system. Under hydroponic culture conditions, the ADF, NDF, RFV,

and yield of WL903 ranged from 22.77% to 25.75%, 28.31% to 32.42%, 197.46 to 233.84 (unitless), and 0.41 kg·m$^{-2}$ to 0.59 kg·m$^{-2}$, respectively. For countries with limited arable land, hydroponics can be another viable way to grow alfalfa. These results provide a reference for the selection of alfalfa varieties in hydroponic culture. However, the specificity of the study conditions should be noted. Future research should focus on nutrient solution optimization, environmental adaptability expansion, and large-scale cost evaluation to promote the practical application of alfalfa hydroponic technology.

## Supporting information

**S1 Table. Complete data information, contains all the data used for data analysis.**
(PDF)

**S1 Fig. Yield and leaf-to-stem ratio of different alfalfa varieties over six consecutive harvests,yield map.**
(TIF)

**S2 Fig. Yield and leaf-to-stem ratio of different alfalfa varieties over six consecutive harvests,leaf stem ratio diagram.**
(TIF)

**S3 Fig. Nutritional quality of different alfalfa varieties over six consecutive harvests,ADF content diagram.**
(TIF)

**S4 Fig. Nutritional quality of different alfalfa varieties over six consecutive harvests,NDF content diagram.**
(TIF)

**S5 Fig. Nutritional quality of different alfalfa varieties over six consecutive harvests,RFV content diagram.**
(TIF)

**S6 Fig. Nutritional quality of different alfalfa varieties over six consecutive harvests,CP content diagram.**
(TIF)

**S7 Fig. Analysis of relationship,Correlation coefficient matrix between yield and nutritional quality traits of alfalfa.**
(TIF)

**S8 Fig. principal component analysis,principal component analysis diagram.**
(TIF)

## Acknowledgments

Thank you very much to the Institute of Grassland Research of CAAS for providing the site for the experiment. I thank Meiying Liu and Feng Li for their guidance and help in the process of writing the article.Thank you Wenlong Li for your help during this test.

## Author contributions

**Conceptualization:** Meiying Liu, Yang Bai, Feng Li.

**Data curation:** Yang Bai.

**Funding acquisition:** Meiying Liu, Feng Li.

**Investigation:** Meiying Liu, Yang Bai, Feng Li, Wenlong Li.

**Methodology:** Yang Bai.

**Project administration:** Meiying Liu, Yang Bai, Feng Li, Wenlong Li.

**Supervision:** Meiying Liu, Feng Li.

**Writing – original draft:** Yang Bai.

**Writing – review & editing:** Yang Bai.

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
