## [Decision Letter · Decision Letter 0]

14 Aug 2025

Dear Dr.  Liu,

Thank you for submitting your manuscript to PLOS ONE. After careful consideration, we feel that it has merit but does not fully meet PLOS ONE’s publication criteria as it currently stands. Therefore, we invite you to submit a revised version of the manuscript that addresses the points raised during the review process.

We look forward to receiving your revised manuscript.

Kind regards,

Durgesh Kumar Jaiswal, Ph.D.

Academic Editor

PLOS ONE

Journal Requirements:

“This work was supported by the Central Public-intercst Soentll cInstitution Basal Research Fund(No.Y2025CY21),National Dairy Technology Innovation Center Project ( Project No. : 2022SR-1 ) ; hohhot Science and Technology Innovation Talent Project ( Project No. : 2022RC-IRI-2 ).”

4. We note that your Data Availability Statement is currently as follows: All relevant data are within the manuscript and in Supporting Information files.

6. We note you have included a table to which you do not refer in the text of your manuscript. Please ensure that you refer to Table 1 in your text; if accepted, production will need this reference to link the reader to the Table.

Additional Editor Comments:

Thank you for your submission on hydroponic alfalfa production. While the topic is timely and of potential value, the manuscript has major shortcomings in study purpose clarity, methodological detail, experimental design, statistical analysis, and consistency of data presentation. Key methodological information is missing, several results are ambiguously reported or misinterpreted, and statistical approaches require reconsideration. Please address all points outlined in the detailed reviewer comments to ensure the study meets publication standards.

Reviewers' comments:

Reviewer's Responses to Questions

**Comments to the Author**

1. Is the manuscript technically sound, and do the data support the conclusions?

Reviewer #1: No

Reviewer #2: Partly

Reviewer #3: No

2. Has the statistical analysis been performed appropriately and rigorously?

Reviewer #1: N/A

Reviewer #2: Yes

Reviewer #3: No

3. Have the authors made all data underlying the findings in their manuscript fully available?

Reviewer #1: Yes

Reviewer #2: Yes

Reviewer #3: No

4. Is the manuscript presented in an intelligible fashion and written in standard English?

Reviewer #1: Yes

Reviewer #2: Yes

Reviewer #3: No

Reviewer #1: Recently, studies on hydroponic farming methods have become quite popular among researchers. . Although I think that such studies should be done, I think that the issue I mentioned below is the biggest handicap of this article.

The purpose of the study is unclear. Is the purpose of the study the possibilities of growing alfalfa in hydroponic environment or the effectiveness of the hydroponic medium used?

The rationale for the proposed study is not clear and valid. The protocol is not technically sound. The methodology is not feasible and detailed enough to make the work replicable

The analysis of the changes occurred during the harvesting periods of alfalfa in hydroponic medium is extremely important because it can contribute to the optimization of the obtaining cultivation. However, more attention needs to be paid to experimental design so that the relevance of the data can be justified and applicable at the agricultural level. In the presented study there is no clear justification of the reason for choosing the hydroponic medium, therefore it would be useful to complete the motivation.

It is seen that the alfalfa in the study has higher protein content and lower NDF content compared to those grown in the field. This may be due to the Hoagland nutrient solution grown, the content of this solution should be given.

The experimental design should be explained. Six cuttings were made, but the interval between cuttings and growing conditions are not given.

High levels of concentrate feed (such as alfalfa) in the ration cause digestive disorders, bloat and metabolic diseases such as acidosis in animals. Therefore, forage crops such as alfalfa are planted in mixtures and added to the ration. It should be stated what kind of suggestion hydroponic cultivation brings to this subject.

Reviewer #2: 1-The purpose of the study should be emphasized at the end of the introduction. This study should be distinguished from other studies. However, this has not been done in this study.

2-The terms “variety” and “cultivar” are used interchangeably in some places. One of these terms should be chosen.

3-The methods section is quite inadequate. Where was the experiment conducted? When was it conducted? When was the harvest carried out? What is the composition of the nutrient solution provided? Answers to many such questions are not found in the materials and methods section.

4-In the tables, varieties are numbered as A, B, C, etc., but in the text, variety names are used instead of numbers, which makes it difficult to understand the text.

5-Although there is no parameter called DDM in Table 2, it has been interpreted in the text.

6-Although the unit of the DMI parameter is %, it has not been used in the text.

7-A protein content of 30% or higher in alfalfa is not considered normal. Such results can be obtained in very early stages. Alfalfa is generally harvested at the 10% flowering stage, and during this period, the crude protein content ranges between 15% and 25%. How often was the harvest conducted?

8-It is incorrect to present forage yield as a percentage in Table 2.

9-Relationship analysis and PCA analysis were conducted in the text. However, no comments were made about these in the results section.

Alfalfa cultivation under hydrponic conditions has been considered a topical issue. However, for the reasons mentioned above, it was not deemed appropriate to publish this study in its current form.

After the necessary corrections have been made, it would be more appropriate to have the article re-evaluated (by other reviewers, if necessary).

Reviewer #3: General Comments

This manuscript examines the impact of six mowing cycles on the yield and nutritional quality of five alfalfa varieties (Zhongcao No. 13, WL440HQ, WL525HQ, WL903, and WL712) under hydroponic conditions. Employing principal component analysis (PCA) and membership function analysis, the study identifies WL903 as superior, with a yield of 4888.31 Kg·hm⁻² and nutritional quality metrics (e.g., 24.37% ADF, 29.92% NDF, 217.94 RFV). The research is novel, addressing a gap in the literature by comparing multiple cultivars across six cuttings in a hydroponic system, which is highly relevant for sustainable forage production in resource-constrained regions like China. However, the manuscript has critical flaws that render it unsuitable for publication in PLOS ONE in its current form. These include an incomplete introduction, misrepresentation of “cutting frequency,” inappropriate statistical design, failure to address significant interaction effects, inadequate methodological detail, inconsistent terminology, limited engagement with prior research, and overstated conclusions. Below, I outline the major and minor issues, and recommendations for improvement.

Major issues

Incomplete introduction

The introduction fails to connect the background on alfalfa and hydroponics to the study’s focus. It discusses global alfalfa production and China’s import trends (2008–2016) but does not link these to hydroponic cultivation or the study’s objectives. Additionally, no objectives or hypotheses are stated, leaving readers unclear about the study’s purpose and expected outcomes. This violates PLOS ONE’s requirement for a clear rationale and objective, hindering assessment of the study’s relevance and novelty. Recommendation: Revise the introduction to link the background to hydroponic alfalfa cultivation, emphasizing its relevance to China’s forage needs. Clearly state the objectives (e.g., to compare yield and quality of five alfalfa varieties across six consecutive cuttings under hydroponic conditions) and hypotheses (e.g., specific cultivars will outperform others due to genetic differences, or quality will vary across cuttings).

Misrepresentation of “cutting frequency” and inappropriate statistical design

The manuscript refers to “cutting frequency” as a factor but does not test different frequencies of cutting (e.g., cutting twice vs. six times over a fixed period). Instead, it examines the evolution of response variables (yield, nutritional quality, leaf-to-stem ratio) over six consecutive cuttings applied uniformly to all cultivars. The experimental design is presented as a factorial design, analysed with two-way ANOVA, but this is inappropriate because the cuttings are repeated measures on the same experimental units (plants), not independent units for each cutting event. A repeated measures analysis is needed to account for the correlation between consecutive cuttings on the same subjects. The misuse of “cutting frequency” misrepresents the study’s design, and the factorial ANOVA violates statistical assumptions, potentially leading to invalid conclusions about main and interaction effects. This undermines the study’s credibility and interpretability. Recommendation: Rename “cutting frequency” to “consecutive cuttings” or “cutting cycles” to accurately reflect the design. Reanalyse the data using a repeated measures ANOVA or a mixed-effects model, treating cuttings as a within-subject factor and varieties as a between-subject factor. Report the model structure, including covariance assumptions, and update results and discussion to reflect the repeated measures analysis.

Failure to address significant interaction effects

Table 2 indicates significant variety-by-cutting interaction effects (P < 0.001) for all nutritional variables (ADF, NDF, CP, RFV, DMI) and yield. DDM is not presented in table 2! However, the results and discussion focus only on main effects (variety and cutting), ignoring how specific combinations of variety and cutting influence outcomes (e.g., WL903’s performance across cuttings). Additionally, no p-values are reported for main or interaction effects for the leaf-to-stem ratio, despite its discussion (Fig 1a). Ignoring significant interactions misrepresents the study’s findings, as the effect of cutting depends on the variety. The lack of statistical analysis for the leaf-to-stem ratio further reduces rigor. Recommendation: Analyse and discuss variety-by-cutting interaction effects using post-hoc tests or interaction plots to clarify specific treatment combinations. Report p-values and statistical details for the leaf-to-stem ratio, ensuring consistency with other variables.

Inconsistent terminology and labelling

The manuscript inconsistently labels varieties as A, B, C, D, and E in the experimental design section while using their names (e.g., Zhongcao No. 13, WL903) elsewhere, creating ambiguity. This reduces clarity and undermines confidence in data presentation. Recommendation: Use variety names consistently throughout the text, tables, and figures, removing A–E labels or defining their correspondence explicitly.

Inadequate methodological detail

The experimental design lacks details critical for reproducibility, including:

- Hydroponic system type (e.g., Nutrient Film Technique, Deep Water Culture).

- Rationale for selecting the five cultivars or their genetic/agronomic traits.

- Light treatment specifics (3.5 h/3.5 h/3 h with 10 h intervals), including intensity, source, and whether it represents light/dark cycles.

- Nutrient solution details (e.g., composition, pH, electrical conductivity for 1.5x Hoagland solution).

- Sample size (e.g., replicates per variety per cutting), plant density, and physical setup (e.g., tray size, growing area).

These omissions prevent replication and assessment of robustness, violating PLOS ONE’s standards. Recommendation: Provide a comprehensive description of the hydroponic system, nutrient solution, light treatment, plant density, and replication. Justify cultivar selection based on genetic or agronomic traits. Clarify the light schedule’s relevance to alfalfa growth.

Questionable data reporting and analysis

- Yield units: The use of Kg·hm⁻² (e.g., 4888.31 Kg·hm⁻² for WL903) is inappropriate for hydroponic systems, which typically use kg·m⁻² due to smaller growing areas.

- Statistical transparency: The two-way ANOVA (inappropriate analysis) and Tukey’s test are reported, but degrees of freedom, F-values, and exact P-values are missing. PCA results lack loading values, and the membership function methodology (e.g., D-value calculations) is not explained.

These issues reduce transparency and interpretability. Recommendation: Justify or convert yield units to kg·m⁻². Include detailed statistical outputs (e.g., Repeated Measurements ANOVA tables, PCA loadings) in the text or supplementary materials. Describe the membership function methodology, including equations and assumptions.

Limited engagement with prior research

The manuscript cites relevant studies (e.g., Feedipedia, Al-Karaki and Al-Momani, 2011) but does not compare its findings to prior hydroponic alfalfa research. The novelty of six cuttings is not contextualised. This limits the study’s ability to demonstrate its contribution. Recommendation: Compare yield, nutritional quality, and cutting effects with prior hydroponic alfalfa studies. Highlight the novelty of the multi-cultivar, six-cutting approach.

Overstated conclusions and lack of limitations

The conclusions advocate WL903 as the “primary germplasm resource” without discussing limitations, such as system specificity, scalability, or performance under varying conditions. The claim that hydroponics is “feasible” for countries with limited arable land lacks cost or scalability data. This overstates the study’s implications and reduces rigor. Recommendation: Revise conclusions to reflect context-specific findings and add a limitations section addressing system specificity, scalability, and condition variability.

Minor Issues

Language and clarity

Awkward phrasing and grammatical errors (e.g., “alfalfa, also known as alfalfa,” “stubble times” instead of “cutting cycles,” “super alfalfa” as an informal term) hinder readability. Conduct a thorough language edit, using scientific terms and replacing vague or informal terms.

Introduction context

The introduction’s focus on China’s alfalfa imports trends (2008–2016) is tangential and not linked to hydroponics. Streamline the introduction to emphasise hydroponic alfalfa research and its relevance to China’s forage needs.

Citation formatting

Citations are inconsistent, and the reference list is not provided, preventing source verification. Provide a complete reference list in PLOS ONE’s format and ensure consistent in-text citations.

Tables and figures

Tables titles and figures captions should be self-explanatory, i.e., readers should not need to screen through the main text to understand what is reported in the tables.

Use Table 2 to present the Repeated Measurements ANOVA (degrees of freedom, F-values, and exact P-values) for the treatment factor (variety) and the time effect factor (cutting number) for all the variables: ADF, NDF, CP, RFV, DMI, DDM, leaf-to-stem ratio and yield Retain figures 1 and 2, which present the means of the combination of all the combinations of variety by cutting number, but assign the letters comparing the 30 different means whenever the interaction effect variety by cut is significant.

Recommendation

The manuscript addresses a novel and timely research question but is unsuitable for publication in PLOS ONE due to critical flaws: an incomplete introduction, misrepresentation of “cutting frequency,” inappropriate use of factorial ANOVA instead of repeated measures analysis, failure to address significant variety-by-cutting interaction effects, inadequate methodological detail, inconsistent terminology, limited engagement with prior research, and overstated conclusions.

.

Reviewer #1: No

Reviewer #2: **Yes:**Dr. Erdal CacanDr. Erdal CacanDr. Erdal CacanDr. Erdal Cacan

Reviewer #3: **Yes:**Jordana RiveroJordana RiveroJordana RiveroJordana Rivero

---

## [Author Response · Author response to Decision Letter 1]

26 Sep 2025

Response to Editor

Dear Durgesh Kumar Jaiswal,

We sincerely appreciate your letter and the reviewers' insightful and constructive comments on our manuscript entitled “Effects of harvest number on the yield and quality of different alfalfa varieties under hydroponic conditions ” Their valuable input has significantly contributed to the refinement of our manuscript and offered critical guidance for our future research endeavors. Following a thorough analysis of the reviewers' feedback, we have meticulously prepared a comprehensive, point-by-point response.

1.We have meticulously revised our manuscript in preparation for resubmission.

2.We have addressed the financial disclosure issues. Specifically, this work was supported by the Central Public-interest Scientific Institution Basal Research Fund (Y2025CY21), the National Dairy Technology Innovation Center Project (2022SR-1), the “Technology Xingmeng” Action Key Projects (NMKJXM202307-01), and the Project of Technology Plan in Inner Mongolia (2025YFHH0270). The funding sources provided the research facilities and equipment necessary for this study.

3.We have added a second author, Wenlong Li, who contributed to data collection during the thesis development.

4.We have thoroughly revised the manuscript in response to the reviewers' comments and have provided well-reasoned, point-by-point responses to each reviewer’s feedback.

5.All modifications in the revised manuscript have been highlighted in yellow, with precise references to the corresponding reviewer comments for ease of review.

We sincerely appreciate the opportunity to improve our manuscript and look forward to the possibility of its publication in PLOS ONE.

Best regards.

Response to reviews

Reviewer 1

Comment 1:The purpose of the study is unclear. Is the purpose of the study the possibilities of growing alfalfa in hydroponic environment or the effectiveness of the hydroponic medium used?

Response: Existing research predominantly focuses on single-variety or single-harvest evaluations, lacking systematic comparisons of multiple varieties under high-frequency harvest regimes. This study addresses this gap by systematically evaluating five mainstream alfalfa varieties across six consecutive hydroponic harvests. The objective is to compare their yield and quality performance under controlled hydroponic conditions, thereby identifying elite varieties best suited for hydroponic cultivation. The findings provide practical guidance for forage production in resource-limited regions.

Comment 2:The rationale for the proposed study is not clear and valid. The protocol is not technically sound. The methodology is not feasible and detailed enough to make the work replicable

Response: The experimental design has been optimized to enhance technical stringency, and the research methods have been described more precisely to guarantee replicability of the work. We thank the reviewer for their feedback.

Comment 3:The analysis of the changes occurred during the harvesting periods of alfalfa in hydroponic medium is extremely important because it can contribute to the optimization of the obtaining cultivation. However, more attention needs to be paid to experimental design so that the relevance of the data can be justified and applicable at the agricultural level. In the presented study there is no clear justification of the reason for choosing the hydroponic medium, therefore it would be useful to complete the motivation.

Response: Thank you for the correction. The introduction has been expanded to explicitly justify the choice of hydroponics, and the study’s objectives are now more clearly articulated in the concluding remarks.

Comment 4:It is seen that the alfalfa in the study has higher protein content and lower NDF content compared to those grown in the field. This may be due to the Hoagland nutrient solution grown, the content of this solution should be given.

Response: Thank you for the correction. The detailed formulation of the Hoagland nutrient solution has been incorporated into the experimental design section.

Comment 5:The experimental design should be explained. Six cuttings were made, but the interval between cuttings and growing conditions are not given.

Response: Thank you for the observation. The experimental design has been revised to include a comprehensive description of the six consecutive harvests, including harvesting intervals and growth conditions.

Comment 6:High levels of concentrate feed (such as alfalfa) in the ration cause digestive disorders, bloat and metabolic diseases such as acidosis in animals. Therefore, forage crops such as alfalfa are planted in mixtures and added to the ration. It should be stated what kind of suggestion hydroponic cultivation brings to this subject.

Response: Thank you for this valuable perspective. The discussion section has been expanded to include recommendations on the dietary incorporation of hydroponically grown alfalfa, addressing potential digestive and metabolic risks.

Reviewer 2

Comment 1:The purpose of the study should be emphasized at the end of the introduction. This study should be distinguished from other studies. However, this has not been done in this study.

Response: Thank you for highlighting this issue. We have revised the introduction to explicitly state the research objective: While existing studies primarily focus on single varieties or single-harvest scenarios, there remains a lack of systematic comparisons across multiple varieties under high-frequency harvesting conditions. In this study, we evaluated five mainstream alfalfa varieties subjected to six consecutive hydroponic harvests to compare their yield and quality performance.

Comment 2:The terms “variety” and “cultivar” are used interchangeably in some places. One of these terms should be chosen.

Response: Thank you for this observation. We have unified the terminology throughout the manuscript, exclusively using the term “variety.”

Comment 3:The methods section is quite inadequate. Where was the experiment conducted? When was it conducted? When was the harvest carried out? What is the composition of the nutrient solution provided? Answers to many such questions are not found in the materials and methods section.

Response: Thank you for your feedback. We have significantly expanded the Materials and Methods section to provide a comprehensive description of the experimental design, including the abovementioned details.

Comment 4:In the tables, varieties are numbered as A, B, C, etc., but in the text, variety names are used instead of numbers, which makes it difficult to understand the text.

Response: Thank you for pointing this out. We have revised the manuscript to ensure consistent use of variety names (rather than alphanumeric codes) throughout the text and tables.

Comment 5:Although there is no parameter called DDM in Table 2, it has been interpreted in the text.

Response: Thank you for catching this discrepancy. We have now included the specific DDM analysis results in Table 2 for clarity.

Comment 6:Although the unit of the DMI parameter is %, it has not been used in the text.

Response: Thank you for this correction. We have ensured that all DMI values in the text are presented with the correct % unit.

Comment 7:A protein content of 30% or higher in alfalfa is not considered normal. Such results can be obtained in very early stages. Alfalfa is generally harvested at the 10% flowering stage, and during this period, the crude protein content ranges between 15% and 25%. How often was the harvest conducted?

Response: Thank you for this insightful question. The alfalfa in this study was harvested every 30 days, for a total of six consecutive harvests.

Comment 8:It is incorrect to present forage yield as a percentage in Table 2.

Response:Thanks for your guidance, the units of output in Table 2 have been modified.

Comment 9:Relationship analysis and PCA analysis were conducted in the text. However, no comments were made about these in the results section.

Response: Thank you for this suggestion. We have expanded the Discussion section to include a detailed interpretation of the relationship and PCA analyses.

Reviewer 3

Comment 1:Incomplete introduction

The introduction fails to connect the background on alfalfa and hydroponics to the study’s focus. It discusses global alfalfa production and China’s import trends (2008–2016) but does not link these to hydroponic cultivation or the study’s objectives. Additionally, no objectives or hypotheses are stated, leaving readers unclear about the study’s purpose and expected outcomes. This violates PLOS ONE’s requirement for a clear rationale and objective, hindering assessment of the study’s relevance and novelty. Recommendation: Revise the introduction to link the background to hydroponic alfalfa cultivation, emphasizing its relevance to China’s forage needs. Clearly state the objectives (e.g., to compare yield and quality of five alfalfa varieties across six consecutive cuttings under hydroponic conditions) and hypotheses (e.g., specific cultivars will outperform others due to genetic differences, or quality will vary across cuttings).

Response: We appreciate your precious feedback. The introduction has been revised to clarify the study’s purpose and establish the connection between the background and hydroponic alfalfa cultivation.

Comment 2:Misrepresentation of “cutting frequency” and inappropriate statistical design

The manuscript refers to “cutting frequency” as a factor but does not test different frequencies of cutting (e.g., cutting twice vs. six times over a fixed period). Instead, it examines the evolution of response variables (yield, nutritional quality, leaf-to-stem ratio) over six consecutive cuttings applied uniformly to all cultivars. The experimental design is presented as a factorial design, analysed with two-way ANOVA, but this is inappropriate because the cuttings are repeated measures on the same experimental units (plants), not independent units for each cutting event. A repeated measures analysis is needed to account for the correlation between consecutive cuttings on the same subjects. The misuse of “cutting frequency” misrepresents the study’s design, and the factorial ANOVA violates statistical assumptions, potentially leading to invalid conclusions about main and interaction effects. This undermines the study’s credibility and interpretability. Recommendation: Rename “cutting frequency” to “consecutive cuttings” or “cutting cycles” to accurately reflect the design. Reanalyse the data using a repeated measures ANOVA or a mixed-effects model, treating cuttings as a within-subject factor and varieties as a between-subject factor. Report the model structure, including covariance assumptions, and update results and discussion to reflect the repeated measures analysis.

Response: Thank you for your valuable suggestion. The term “cutting frequency” has been replaced with “continuous harvesting,” and the original two-way ANOVA (Table 2) has been supplanted by a repeated measures ANOVA approach.

Comment 3:Failure to address significant interaction effects

Table 2 indicates significant variety-by-cutting interaction effects (P < 0.001) for all nutritional variables (ADF, NDF, CP, RFV, DMI) and yield. DDM is not presented in table 2! However, the results and discussion focus only on main effects (variety and cutting), ignoring how specific combinations of variety and cutting influence outcomes (e.g., WL903’s performance across cuttings). Additionally, no p-values are reported for main or interaction effects for the leaf-to-stem ratio, despite its discussion (Fig 1a). Ignoring significant interactions misrepresents the study’s findings, as the effect of cutting depends on the variety. The lack of statistical analysis for the leaf-to-stem ratio further reduces rigor. Recommendation: Analyse and discuss variety-by-cutting interaction effects using post-hoc tests or interaction plots to clarify specific treatment combinations. Report p-values and statistical details for the leaf-to-stem ratio, ensuring consistency with other variables.

Response: Thank you for your constructive feedback. The leaf-to-stem ratio has been incorporated into the repeated measures ANOVA (Table 2), and the significance of differences between Fig. 1 and Fig. 2 has been annotated.

Comment 4:Inconsistent terminology and labelling

The manuscript inconsistently labels varieties as A, B, C, D, and E in the experimental design section while using their names (e.g., Zhongcao No. 13, WL903) elsewhere, creating ambiguity. This reduces clarity and undermines confidence in data presentation. Recommendation: Use variety names consistently throughout the text, tables, and figures, removing A–E labels or defining their correspondence explicitly.

Response: Thank you for pointing this out. We have removed all A–E labels from the manuscript and replaced them with the corresponding cultivar names.

Comment 5:Inadequate methodological detail

The experimental design lacks details critical for reproducibility, including:

- Hydroponic system type (e.g., Nutrient Film Technique, Deep Water Culture).

- Rationale for selecting the five cultivars or their genetic/agronomic traits.

- Light treatment specifics (3.5 h/3.5 h/3 h with 10 h intervals), including intensity, source, and whether it represents light/dark cycles.

- Nutrient solution details (e.g., composition, pH, electrical conductivity for 1.5x Hoagland solution).

- Sample size (e.g., replicates per variety per cutting), plant density, and physical setup (e.g., tray size, growing area).These omissions prevent replication and assessment of robustness, violating PLOS ONE’s standards. Recommendation: Provide a comprehensive description of the hydroponic system, nutrient solution, light treatment, plant density, and replication. Justify cultivar selection based on genetic or agronomic traits. Clarify the light schedule’s relevance to alfalfa growth.

Response: Thank you for your constructive feedback. We have now included detailed descriptions of the experimental design, addressing all the aforementioned methodological aspects.

Comment 6:Questionable data reporting and analysis

- Yield units: The use of Kg·hm⁻² (e.g., 4888.31 Kg·hm⁻² for WL903) is inappropriate for hydroponic systems, which typically use kg·m⁻² due to smaller growing areas.

- Statistical transparency: The two-way ANOVA (inappropriate analysis) and Tukey’s test are reported, but degrees of freedom, F-values, and exact P-values are missing. PCA results lack loading values, and the membership function methodology (e.g., D-value calculations) is not explained.

These issues reduce transparency and interpretability. Recommendation: Justify or convert yield units to kg·m⁻². Include detailed statistical outputs (e.g., Repeated Measurements ANOVA tables, PCA loadings) in the text or supplementary materials. Describe the membership function methodology, including equations and assumptions.

Response: Thank you for your valuable comments. We have revised the yield units to kg·m⁻² and included the PCA loading matrix (Table 3). Additionally, the calculation method for the membership function has been elaborated in the data statistics and analysis section.

Comment 7:Limited engagement with prior research

The manuscript cites relevant studies (e.g., Feedipedia, Al-Karaki and Al-Momani, 2011) but does not compare its findings to prior hydroponic alfalfa research. The novelty of six cuttings is not contextualised. This limits the study’s ability to demonstrate its contribution. Recommendation: Compare yield, nutritional quality, and cutting effects with prior hydroponic alfalfa studies. Highlight the novelty of the multi-cultivar, six-cutting approach.

Response: We sincerely appreciate these insightful comments. The Discussion section has been enhanced with additional references (e.g., Deng R, Xiang QH, and Zhang DH, 2014) to strengthen the comparative analysis between our findings and established hydroponic alfalfa research.

Comment 8:Overstated conclusions and lack of limitations

The conclusions advocate WL903 as the “primary germplasm resource” without discussing limitations, such as system

---

## [Decision Letter · Decision Letter 1]

29 Dec 2025

Dear Dr. Liu,

Thank you for submitting your manuscript to PLOS ONE. After careful consideration, we feel that it has merit but does not fully meet PLOS ONE’s publication criteria as it currently stands. Therefore, we invite you to submit a revised version of the manuscript that addresses the points raised during the review process.

We look forward to receiving your revised manuscript.

Kind regards,

Debasis Mitra

Academic Editor

PLOS One

Journal Requirements:

Reviewers' comments:

Reviewer's Responses to Questions

**Comments to the Author**

Reviewer #2: All comments have been addressed

Reviewer #4: (No Response)

2. Is the manuscript technically sound, and do the data support the conclusions?

Reviewer #2: Yes

Reviewer #4: Yes

3. Has the statistical analysis been performed appropriately and rigorously?

Reviewer #2: N/A

Reviewer #4: Yes

4. Have the authors made all data underlying the findings in their manuscript fully available?

Reviewer #2: Yes

Reviewer #4: Yes

5. Is the manuscript presented in an intelligible fashion and written in standard English?

Reviewer #2: Yes

Reviewer #4: Yes

Reviewer #2: 1- The RFV unit is given in the table as a percentage. RFV is unitless.

2- It is uncommon to see both "ABC" and "abc" lettering appearing together in tables. A change may be necessary at this point.

3- The summary could be further detailed.

Reviewer #4: Based on which parameter did you select five varieties of alfa-alfa? i.e yield or production, quality, local variety, resistance? Please describe it with a sentence to the experimental design paragraph.

Did you follow a protocol to establish 10 h light for the experiment? Please insert a reference or if this procedure is yours, please indicate within the text.

Fig 1, Fig 2 and Fig 3 please make them larger in order to read data inside the graphs.

Please describe which were principal components that you have analyzed? i.e yield, quality, production

.

Reviewer #2: No

Reviewer #4: No

---

## [Author Response · Author response to Decision Letter 2]

14 Jan 2026

Response to Editor

Dear Debasis Mitra,

We sincerely appreciate your letter and the reviewers' insightful and constructive comments on our manuscript entitled “Effects of harvest number on the yield and quality of different alfalfa varieties under hydroponic conditions ” Their valuable input has significantly contributed to the refinement of our manuscript and offered critical guidance for our future research endeavors. Following a thorough analysis of the reviewers' feedback, we have meticulously prepared a comprehensive, point-by-point response.

1.We have meticulously revised our manuscript in preparation for resubmission.

2.We have thoroughly revised the manuscript in response to the reviewers' comments and have provided well-reasoned, point-by-point responses to each reviewer’s feedback.

3.All modifications in the revised manuscript have been highlighted in yellow, with precise references to the corresponding reviewer comments for ease of review.

We sincerely appreciate the opportunity to improve our manuscript and look forward to the possibility of its publication in PLOS ONE.

Best regards.

Response to reviews

Reviewer 1

Comment 1:The RFV unit is given in the table as a percentage. RFV is unitless.

Response:Thank you for your suggestion. I have removed the unit for RFV in the tables.

Comment 2:It is uncommon to see both "ABC" and "abc" lettering appearing together in tables. A change may be necessary at this point.

Response:Thank you very much for this detailed comment, which has led us to a clearer description of the statistical presentation of the table. The use of both lowercase and uppercase letters in our tables is not a format error, but a standard and necessary presentation determined by the repeated measures analysis of variance method used in this study. In this design, “variety” is an inter-group factor, and “harvest time” is an intra-group factor ( repeated measurement ). Therefore, data analysis will naturally produce two different types of post-hoc comparisons : 1.For intra-group factors ( harvest ) : We compared the differences in different harvest times ( harvest ) within the same variety. The lowercase letters ( a, b, c... ) in the table represent this vertical, time-series comparison. 2.For between-group factors ( varieties ) : We compared the differences between different varieties under a given stubble. The uppercase letters ( A, B, C... ) in the table represent this horizontal, treatment-group comparison. Nevertheless, we fully understand your original intention. In order to make readers understand the table more clearly, the analysis method has been explained in the notes.

Comment 3:The summary could be further detailed.

Response:Your point is indeed crucial. I have revised the abstract section accordingly, providing a more detailed description of the experimental content and results.

Comment 4:Based on which parameter did you select five varieties of alfa-alfa? i.e yield or production, quality, local variety, resistance? Please describe it with a sentence to the experimental design paragraph.

Did you follow a protocol to establish 10 h light for the experiment? Please insert a reference or if this procedure is yours, please indicate within the text.

Fig 1, Fig 2 and Fig 3 please make them larger in order to read data inside the graphs.

Please describe which were principal components that you have analyzed? i.e yield, quality, production

Response:Thank you for your very useful comments. I have explained the reasons for choosing these five alfalfa varieties in the experimental design section and explained the origin of the lighting time design. I have enlarged the picture. In the results and discussion section of principal component analysis, the specific analysis of which indicators is described in detail.

---

## [Decision Letter · Decision Letter 2]

19 Mar 2026

Effects of harvest number on the yield and quality of different alfalfa varieties under hydroponic conditions

PONE-D-25-32806R2

Dear Dr. Meiying Liu,

We’re pleased to inform you that your manuscript has been judged scientifically suitable for publication and will be formally accepted for publication once it meets all outstanding technical requirements.

Kind regards,

Andrey Nagdalian

Academic Editor

PLOS One

Additional Editor Comments (optional):

Reviewers' comments:

Reviewer's Responses to Questions

**Comments to the Author**

Reviewer #2: All comments have been addressed

Reviewer #4: All comments have been addressed

2. Is the manuscript technically sound, and do the data support the conclusions?

Reviewer #2: Yes

Reviewer #4: Yes

3. Has the statistical analysis been performed appropriately and rigorously?

Reviewer #2: Yes

Reviewer #4: Yes

4. Have the authors made all data underlying the findings in their manuscript fully available?

Reviewer #2: Yes

Reviewer #4: Yes

5. Is the manuscript presented in an intelligible fashion and written in standard English?

Reviewer #2: Yes

Reviewer #4: Yes

Reviewer #2: (No Response)

Reviewer #4: Dear Authors,

I would like to thank you for revising the manuscript carefully and for responding to all the issues raised in the previous review. I have reviewed the revised manuscript along with your response to the reviewers' comments.

The authors have successfully addressed all the issues raised in the review by making all the suggested changes, which have greatly improved the quality and scientific value of the manuscript.

In my opinion, the manuscript has greatly improved and is ready for publication in its current form.

Good luck!

.

Reviewer #2: No

Reviewer #4: No

---

## [Editor Report · Acceptance letter]

PONE-D-25-32806R2

PLOS One

Dear Dr. Liu,

I'm pleased to inform you that your manuscript has been deemed suitable for publication in PLOS One. Congratulations! Your manuscript is now being handed over to our production team.

Kind regards,

on behalf of

Dr. Andrey Nagdalian

Academic Editor

PLOS One